# Time to Acceptance of 3 Days for Papers about COVID-19

**Ádám Kun** [1,2,3,4] 

[1] Evolutionary Systems Research Group, Institute of Evolution, Centre for Ecological Research, Klebelsberg Kuno utca 3, 8237 Tihany, Hungary; kunadam@elte.hu

[2] Parmenides Centre for the Conceptual Foundation of Science, Parmenides Foundation, Kirchplatz 1, 82049 Pullach, Germany

[3] MTA-ELTE Theoretical Biology and Evolutionary Ecology Research Group, Pázmány Péter sétány 1/C, 1117 Budapest, Hungary

[4] Institute for Advanced Studies Kőszeg, Chernel utca 14, 9730 Kőszeg, Hungary

**Abstract:** Time to acceptance from submission and time to publication (publication lag) determines how quickly novel information is made available to other scientists and experts. In the medical field, the review process and revisions usually takes 3–4 months; the total time from submission to publication is 8–9 months. During the COVID-19 pandemic, information should be available much faster. The analysis of 833 documents published on SARS-CoV-2 and COVID-19 prior to 19 March 2020 shows that these times shrunk by a factor of ten. The median time to acceptance was three days for all publications, six days for research papers and reviews, four days for case studies and two days for other publication types. The median publication lag was nine days for all publications together, 11 days for research papers, nine days for case studies, 13 days for reviews and seven days for other publications. This demonstrates that the publication process—if necessary—can be sped up. For the sake of scientific accuracy, review times should not be pushed down, but the time from acceptance to actual publication could be shorter.

**Keywords:** coronavirus; COVID-19; SARS-CoV-2; speed of publication; peer review; publication lag; publication efficiency; publication delay

## 1. Introduction

There is an ongoing pandemic caused by a coronavirus named SARS-CoV-2. As of 8 April 2020, 1,496,335 cases had been reported, with 87,617 deaths connected to COVID-19; a month later these number had changed to 3,932,004 cases and 270,905 fatalities. This virus is the seventh coronavirus that can infect humans. HCoV-229E, HCoV-OC43, HCoV-NL63 and HCoV-HKU1 cause seasonal, mild disease (the common cold), and are considered endemic [1]. SARS and MERS have recently jumped from animals to humans [2,3]. SARS is fully contained and eradicated, while MERS is mostly contained, with no significant outbreaks since 2018. In less than two decades, a third novel coronavirus is causing a massive epidemic.

As a novel virus causes a pandemic, there are many unknowns. Scientific research and medical case studies—as well as expert opinions from health professionals—can help others to cope with the situation. Rapid and open communication is key in fighting this pandemic [4], which can be achieved by preprints, speedy publication processes and open access to published findings.

Dissemination can be accelerated by posting finished manuscripts on preprint servers (for example on medrxiv.org). Moreover, preprints are coming out fast [5], which can be beneficial if the information is crucial (and in many cases it is) for practitioners and policy makers. But such manuscripts have

not yet gone through peer-review (which is clearly stated on the preprint's website), and as such it could contain methodological errors and premature conclusions. Meanwhile, the community, i.e., the pool of scientists from whom the peer-reviewers could come, can already begin to discuss the study and point out problems. Thus, in essence, a community peer-review can take place; but it has not replaced traditional peer-review. The status and use of preprints—especially in biomedicine—is still debated [6,7].

Another feature of the dissemination of knowledge on COVID-19 is open access to information. All published papers should be open-access and be available as soon as possible [8]. The Wellcome Trust had issued a call asking publishers to openly share published data and findings. Publishers— particularly the major ones—joined and made all publications on COVID-19 openly accessible.

Fully understanding that speed is important in an unfolding epidemic, we can still ask—how fast can the publication process be? There are studies, mostly in medical sciences, that are relevant here, asking the same question prior to COVID-19. The average time (median) from submission to acceptance was between 87–147 days for medical journals; the mean (median) publication lag, i.e., the time from submission to publication was 99–567 days (Table 1). Production and publication of systematic reviews in the medical field could take up to eight months [9,10] or more than a year [11]. The publication of the results of clinical trials could take years [12–16]. Another study of clinical trials found the median time to acceptance to be 90 days and publication lag to be 124 days [17].

**Table 1.** Time to acceptance and publication lags for medical journals from the literature.

| Journal or Set of Journals | Years | Mean/Median Acceptance Lag (Days) | Mean/Median Total Publication Lag (Days) | Ref. |
|---|---|---|---|---|
| Emergency medicine journals | 1996 | 147 | 208 | [18] |
| *Head Face Medicine* | 2005–2006 | 95.9 | 99.3 | [19] |
| Plastic surgery journals | 2008 | 142.1 | 463.4 | [20] |
| Nursing journals | 2009 | 146 | 262 | [21] |
| Ophthalmology journals | 2010 | 133 | 233 | [22] |
| Original research in otolaryngology journals | 2010–2013 | 123.4 | 220.5 | [23] |
| Case studies in otolaryngology journals | 2010–2013 | 87.1 | 227.9 | [23] |
| Biomedical journals | 2012 | 139.2 | 287.7 | [24] |
| *American Journal of Roentgenology* | 2012 | 115 | 371.1 | [25] |
| Indian medical journals | 2012–2014 | 143.5 | 126–567 | [26] |
| Korean medical journals | 2013–2016 | 102.0 | 246.5 | [27] |
| Journals in general internal medicine and primary care | 2016 | 118 | 242.5 | [28] |
| General medical journals | 2016 | 123 | 224 | [29] |

The above data—while coming from different set of journals and time-frames (publication times had actually decreased somewhat over the years [23])—agree that it takes roughly 3–4 months to be accepted, and yet more to be actually published. With the same timeframe of publication, we would only get a trickling of information from the very beginning of the epidemic in Wuhan. Thus, here we analyze publications from the first 2.5 months of 2020 on COVID-19 to see how fast the publication process had become.

## 2. Materials and Methods

Web of Science was searched for "coronavirus" on the 4th of March 2020, listing only papers published in 2020. Papers dealing with the new coronavirus infection were analyzed and general research on coronaviruses, for example infecting domestic animals or publications on SARS and MERS were omitted. This yielded 85 papers and another 64 not dealing with SARS-CoV-2 were analyzed separately.

To increase the number of publication in our analysis, we also used the COVID-19 open research dataset (CORD-19) [30]. It was downloaded on the 19 March 2020. This was the release available from

13 March 2020. Again, non-English language publications and publication not dealing with the new disease were omitted. In the end, 833 publications were analyzed (Table S1).

There were borderline cases as for inclusion. For example, a review on the detection of different viruses [31] has a few lines on SARS-CoV-2 in the introduction, but fails to mention it as the seventh coronavirus infecting humans later on (listing only six as human infecting). I assume that the line was added to the introduction around last revision / proof phase, but otherwise this publication was in the making before the COVID-19 outbreak. This study is the outlier with 92 days to publication from submission.

We have collected the online publication dates of papers, which are the dates publishers make the articles available on their websites. We did not collect the publication dates of print issues (in case of printed journals), as we were only interested in the earliest availability of the information and not in the general backlog of these journals. If available, the date of submission and acceptance were also recorded. Furthermore, we also collected the latest date the article reported on. In case of case-studies, this marks the end of the study period or in other cases it may indicate the date when writing of the manuscript was finished. Many papers included data on the extent of the pandemic (number of cases and deaths), clearly indicating the date on which such data were collected. If other date was not given, we used this date, however, it is clear from papers with submission, acceptance and publication dates, that many authors updated the report on the current standing of the epidemic in the proof phase, thus we cannot use it to measure the time it took to write/finalize the manuscript.

We also recorded the type of the publication. There were many editorials, news, commentaries, perspectives, viewpoints, highlights published in the past months (these are collectively labeled as other publication). These probably have a different production pipeline than actual research papers. Research papers, case studies and reviews were analyzed separately from the rest of the documents. Each publication was assessed individually, as some letters to the editor or rapid communications are actual case studies/research papers, while others are more like commentaries. Basically, a research study is one having a method section, a case study is the first publication on patients (some of these data also appear in reviews), a review is one labeled as such by the journal; all other articles are considered to belong to some other types.

Please note that the data were collected mostly manually. Most of the publishers' website forbid scripts to parse their contents. In some cases, the required information is only present in pdf form. Data availability can also change even within a journal. The journal *Radiology* (published by the Radiological Society of North America) had full article history up to 12 February 2020, but later papers had only the publication date in the article history. Documents were not selected based on the availability of the above data, all publication meeting our criteria from the employed sources were included.

Where the data were available, we calculated the time from submission (receptance by the editorial office) to acceptance and the time from submission to actual publication (publication lag) separately for research papers, case studies, reviews and the other documents.

To compare these dates to the length of these processes in "pace time", we also analyzed papers having coronavirus in the title, abstract or keyword, which are published in 2020, but not about SARS-CoV-2/COVID-19. Fifty-one documents were identified for which submission date, acceptance date and first publication date are available. These dates may not fall into 2020, but the volume/issue to which they belong were from 2020.

## 3. Results

Among the 833 publications, there are 62 reviews (7.4%), 136 research articles (16.3%), 126 case studies (15.1%) and 509 other publications (61.2%) (editorials, news, commentaries, perspectives, viewpoints, highlights, etc.). Identifiable publication date was found for 97% of the documents (811/833). Publication lag is available for 45% (378/833) of the documents. Time to acceptance, i.e., time between receptance of the manuscript by the journal and acceptance of the study, is available for 44% of the documents (367/833). There is an increasing number of publications (Figure 1) on COVID-19,

albeit our data, unlike the pandemic, do not show exponential increase in the number of documents. The tapering after 2 March is an artefact, as fewer of the published papers got into the database by the timeframe of our investigation.

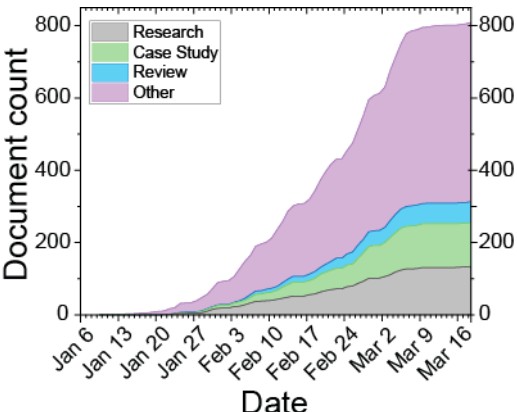

**Figure 1.** Cumulative number of documents by date on COVID-19 in the study period. The different document types are color coded and their count stacked on each other.

The median time to acceptance was 3 days for all publications. In more detail, the median time to acceptance was 6 days for research papers and reviews, 4 days for case studies and 2 days for other publication types (Figure 2). There were publications that were accepted on the same day as they were received by the editorial office. This is probably standard practice for editorials, but seems to be considerably fast for research papers, case studies and reviews. Publications were accepted on the very same day as they were submitted (received by the editorial office) in 5% (5/101) of the research papers, 11% (5/46) of the reviews, 8% (4/53) of the case studies and 23% (38/167) of the other publications. Publication lag was a bit greater, with a median publication lag of 9 days for all publications together. The median publication lag for research papers, case studies, reviews and other publications were 11, 9, 13 and 7 days, respectively (Figure 2). Only one review was published on the same day as it was received. Otherwise, 3 other documents had such record among those with these dates disclosed.

The time it takes the journal (publisher) to actually post the accepted documents online took a median of 4 days. The median lead lag for research papers, case studies, reviews and other publications were 3.5, 5, 6 and 4 days, respectively. This period could measure the speed of the production process, which transforms the accepted manuscript to a published article. But in many cases, provisional pdfs were posted, i.e., the actual production process has not been done yet. Furthermore, journals could also time their article publication.

The speed of publication can be compared to papers on other coronaviruses, including SARS and MERS, appearing in 2020. The 51 documents identified had a median of 91 days from receptance to acceptance and 116 days as publication lag. Minimum, lower quartile, median, upper quartile and maximum are 18, 60, 91, 130 and 240 days for time to acceptance and 22, 77.5, 116, 153 and 241 days for publication lag.

There is one more date that we can analyze during this pandemic. Most of the papers (59%, 488/833) had indicated either the study period (which should be standard) or the extend of the pandemic indicating the exact date on which said data are valid. In 36 cases, this is the same as the time of publication. Surprisingly, in 9 cases the "as of" date is actually later than the first publication of the document, which indicates that it was amended after publication. The time difference is small, median 3 days (1, 1, 1, 2, 3, 3, 4, 5, 19), albeit as seen above, this is comparable to the manuscript review/handling time. One document references a date 19 days after publication. These are not corrections, but still show that documents are updated post publication (not just post acceptance, which includes the proof phase, at which point manuscript can be updated). Among the identified documents, 14 were real

corrections. Albeit all were minor corrections of badly spelled names, mixed up figure references and the like, they are also indications that publications are rushed.

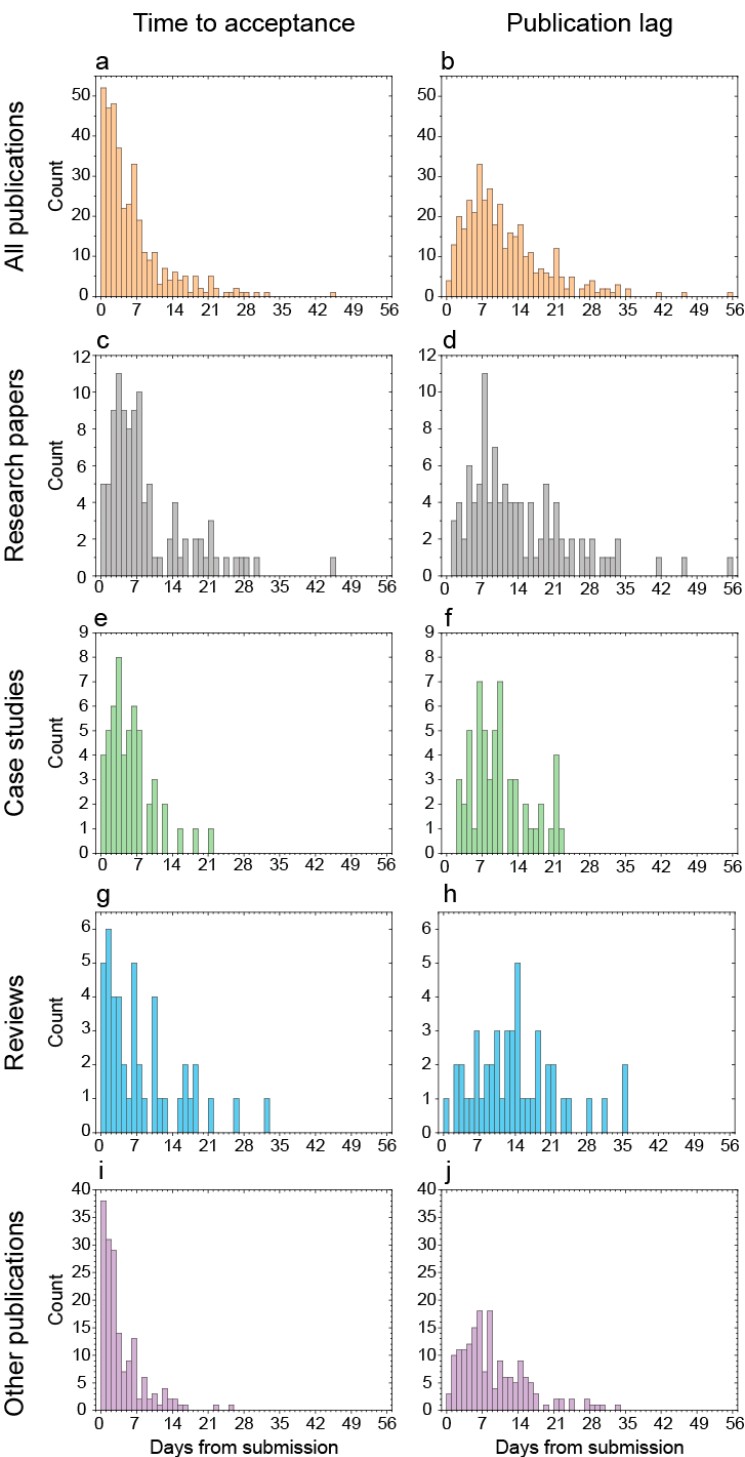

**Figure 2.** Number of documents with given number of days from submission to acceptance (**a**,**c**,**e**,**g**,**i**) and to publication (**b**,**d**,**f**,**h**,**j**). Figures are given together for all publications (**a**,**b**) and for research papers (**c**,**d**), case studies (**e**,**f**), reviews (**g**,**h**) and other document types (editorials, news, commentaries, perspectives, viewpoints, highlights) (**i**,**j**) separately.

## 4. Discussion

Median time to acceptance was 2–6 days depending on publication type. Thus, it can take less than one week to get accepted with COVID-19 related manuscripts. The first milestone in the transformation of a manuscript to a paper is acceptance, at which time scientists may begin to brag about their new achievement. The time from submission to acceptance is quite low for these papers. Acceptance within one days of submission is not unheard of [23], but it is more the exception than the rule. Here, we have found that short review times were not uncommon and even acceptance on the very same day as receptance was observed. Publications on other coronaviruses take three months to review, which fits to the acceptance times reported for other fields/journals referenced in the introduction (Table 1). This time shrunk to its tenth in the recent months for papers on SARS-CoV-two and the pandemic it caused.

Our results need to be taken with a caution though, as papers which may take longer to review would not show up in our analysis because of the short timeframe after the outbreak of the epidemic. While we show that the already published papers were published quickly, there could be studies already submitted by early March, that will be published much later. Consequently, follow-up analyses may uncover longer review times, albeit I think as long as the pandemic lasts there will be a pressure to publish fast with regard to COVID-19. There may be one exception, in which data are withheld because the owners want to make sure their analysis of the data lands in a high impact factor journal [32]. For example, in case of sequences, other scientist can also use them once they are shared with the scientific community, and potentially make the same analysis.

New infectious/transmissible diseases attract considerable attention from the scientific community. Publication on coronaviruses has noticeably increased after the 2002/03 SARS outbreak and the 2012/13 MERS outbreak [33–35]. Ebola, even though known since 1976, got a boost in publication number during the 2015 outbreak [36], and the same is observable for ZIKA after the outbreak in South America in 2015 [37]. The increase in publication output was in all cases considerable, but not as fast as in the case of the COVID-19 pandemic. In case of the ZIKA virus, publication skyrocketed only in 2016 while the presence of the virus was identified in Brazil in the first half of 2015. MERS also attracted modest interest with relatively few articles in the months right after the outbreak, and a peak in 2014, a year after the start of the epidemic [38]. There were only 104 publications (excluding news items) in 2013 [38], which number was already surpassed by COVID-19 publications in the first two months. SARS, on the other hand, caused similar surge of scientific activity as observable for the current epidemic. The virus was identified in February 2003, and it was covered by 6 publications in March, 45 in April, 132 in May and 68 in June [39]. Most the publications in the early months of the SARS pandemic were news items or editorial materials (57%) and only 22% of them were articles [39]. We have found no publication on the speed of the publication process during these past epidemics. Thus, while the similar upwell of publications can be shown, further research is needed to show if publication pace speeds up during global health crises or the quickened publication pace we have identified is a novel phenomenon.

The at least ten times quicker publication process observed for early papers on COIVD-19 requires an explanation. The lead lag, the time it takes accepted manuscripts to be published has approximately been cut in half since the early 2000s, and are around 25 days [40]. Our results show that it can be further lowered, as median acceptance to publication time was found to be 4 days. As indicated earlier, some of the publications have not underwent technical production, but provisional pdfs were already published by journals before the current pandemic. But the shortening of the last, technical part of the publication process could not account for the considerably shorter times from submission to acceptance. Analysis of millions of documents in the 1965–2016 period from all disciplines found that acceptance times have teetered around 100 days [40], with many medical journals having median acceptance delays higher than this figure (Table 1). The time it takes to peer review an article sets the lower limit to the time required for acceptance [41]. The actual peer review by an expert may take only a few hours [42], and thus, the reported acceptance times are entirely feasible. However, it can take considerable time to find that few hours in the usually busy schedule of scientist.

The question is then why journals and reviewers manage to do it quickly now, but not in other cases? One can argue, that in an escalating pandemic, quick information can save lives as practitioners and policy makers can make decisions based on a fuller picture, and accordingly everybody should do its own to help. As an editorial in the medical journal *Thorax* stated, "it is crucial that journals streamline, but maintain high-quality peer review processes" [43]. But we can also argue, that it is true for medical findings in general, and medical journals are not the fastest to publish papers [24].

The fast publication process, which is an important factor in choosing a journal [44], is good for the authors as well as journals. From the journal's perspective the papers can begin to attract citations earlier. Many of the papers on COVID-19 have already attracted citations [35,45,46] which may reflect favorably in this year's impact factor. Journals could pressure authors to quickly write up findings, which can lead to faulty (without any ill intention) results. For example, a study claimed to observe an asymptomatic patient infecting others [47], but it turned out that the patient suppressed symptoms with medication and only appeared to be asymptomatic to outside observers—while feeling ill herself [48]. As one of the authors explained "The need to share information as fast as possible, along with the *New England Journal of Medicine*'s (NEJM) push to publish early, created a lot of pressure" [48]. From the author's perspective, more publications are always welcome, and considering that publication times are painfully slow otherwise [49], some low-hanging fruit should be reaped. The plethora of opinion, perspective and comment papers in various journals point to this direction. There is nothing inherently wrong with the increased number of papers (albeit the general trend of increasing number of papers may not be good for science [50,51]). However, more papers increase the noise-to-information ratio and the faster review process may decrease the reliability of information. Peer-review—be it solicited or community driven—is still the only reliable method of scrutinizing research findings and presentation, and especially amid a global health emergency clear and accurate presentation is key.

This pandemic will change some of our habits. It may also change how scientific information is published and reviewed. The reported fast publication process may be a transient phenomenon only connected to COVID-19 or could bring publication times down across the board as expectation change within the scientific community. Time will tell.

**Supplementary Materials:** The following are available online at http://www.mdpi.com/2304-6775/8/2/30/s1, Table S1: Publication details of COVID19 papers.

**Funding:** This research was funded by National Research, Development and Innovation Office (NKFIH), Grant Number GINOP-2.3.2-15-2016-00057 and by the Institute for Advanced Studies Kőszeg (iASK).

**Conflicts of Interest:** The authors declare no conflict of interest.

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
