# Peer review of "Time to Acceptance of 3 Days for Papers About COVID-19"

_publications, doi:10.3390/publications8020030_

Round 1
Reviewer 1 Report
This article provides excellent data on the significantly truncated time to publication for articles on SARS-COV-2 and COVID-19. The <2 year turnaround time, as compared to the typical 3-4 month review period and 1 year to publication typically expected in medical fields of study, is surprising, even considering the urgent, global circumstances under which these papers are being published.
It would be helpful to know up front what the author believes to be at stake in the presentation of these data. Is it simply that our peer review and academic publication process is too slow, and the current crisis reveals that we can share results that will lead to innovation more quickly? Is it that researchers are taking advantage of an urgent situation to get more publications, which may lead to flaws in methods and misleading findings? Is it that a new peer review model has emerged that we should take advantage of in other contexts?
I believe that the answer to these questions lies in lines 39-41, but the meaning of these sentences is unclear. Is the author going to show us data that reveal that community peer review is currently taking place? Or perhaps show the reader that conditions are ideal for community peer review, but this review is still not taking place? Additionally, does "it" in "It is important" (40) refer to a peer review that should or that is already taking place? Additionally, the statement, "the provided information also needs to be accurate" (40-41) is a given. What will this study reveal to us about the accuracy of the data in the publications studied?
The Discussion section indicates that what is at stake here, or at least what is being criticized, is the opportunist nature of some researchers who are taking advantage of a global crisis to get more publications under their belts, without paying close enough attention to the quality of their research and thereby circulating problematic/flawed/inaccurate research.
If this is not the point of the article, but rather one of many provocations that the author intends to bring up through his study of the turnaround time to publication during the COVID pandemic, it would be helpful for him to list what he believes to be at stake in the Introduction.
I agree that "we need to be aware that in this crisis we not only need information in general, but reliable information" (189-190), but since most (if not all) readers would agree with this premise, it is important to state from the beginning what is at stake here beyond the importance of publishing accurate findings.
Additionally, I would caution against using the language of disease to describe the process of academic publication. A "symptom of a paper" (165) seems insensitive in the middle of a global health crisis.
Author Response
It would be helpful to know up front what the author believes to be at stake in the presentation of these data. Is it simply that our peer review and academic publication process is too slow, and the current crisis reveals that we can share results that will lead to innovation more quickly? Is it that researchers are taking advantage of an urgent situation to get more publications, which may lead to flaws in methods and misleading findings? Is it that a new peer review model has emerged that we should take advantage of in other contexts?
I have considerably rewritten the discussion to be more focused. Indeed, the presented data show that peer-review and the whole publication process can be considerably quicker. But without the pressure to publish, longer publication times are just frustrating. The hype around the pandemic allow some to collect some publications quickly. Furthermore, there is a pressure from granting agencies and institutional leaderships to work on some aspect of the pandemic. The last two newsletter from the umbrella organisation to which my research centre belongs to was about how different institutes contribute to the fight against COVID-19.
Consequently, it would be cheap to blame the authors that they need more papers and some agencies demand research on certain topics. That would manifest in more papers on SARS-CoV-2/COVID-19, but not, in itself, in quicker publication. Only the journal (editors) can sped up the publication process by selecting quicker reviewers (or pressing them to finish their assessment in shorter times) and prioritizing the publication of certain papers. As both the publishers and the authors have an interest in more papers/citations, it is a win-win situation. The only drawback is the potential degradation of the quality of publications.
I believe that the answer to these questions lies in lines 39-41, but the meaning of these sentences is unclear. Is the author going to show us data that reveal that community peer review is currently taking place? Or perhaps show the reader that conditions are ideal for community peer review, but this review is still not taking place? Additionally, does "it" in "It is important" (40) refer to a peer review that should or that is already taking place?
I have added a paragraph on the positive and the negative aspect of preprints. The positive is the more transparent and community driven peer-review. Furthermore, not only the paper can be commented on, but the peer-review also.
Additionally, the statement, "the provided information also needs to be accurate" (40-41) is a given. What will this study reveal to us about the accuracy of the data in the publications studied?
The validity of the data can only be uncovered by further studies, and an analysis of publication times could not do that. But, and that is one of my claim, it can warn the public that the usual rigor of peer-review might have been relaxed.
The Discussion section indicates that what is at stake here, or at least what is being criticized, is the opportunist nature of some researchers who are taking advantage of a global crisis to get more publications under their belts, without paying close enough attention to the quality of their research and thereby circulating problematic/flawed/inaccurate research.
If this is not the point of the article, but rather one of many provocations that the author intends to bring up through his study of the turnaround time to publication during the COVID pandemic, it would be helpful for him to list what he believes to be at stake in the Introduction.
This is exactly my point. But I do not think that all or even the majority of the papers appeared just to have one more line in someone’s publication list. But there could be some, as the topic is “fashionable”. I discuss this in the discussion.
I agree that "we need to be aware that in this crisis we not only need information in general, but reliable information" (189-190), but since most (if not all) readers would agree with this premise, it is important to state from the beginning what is at stake here beyond the importance of publishing accurate findings.
Additionally, I would caution against using the language of disease to describe the process of academic publication. A "symptom of a paper" (165) seems insensitive in the middle of a global health crisis.
As also requested by Reviewer 3, all untasteful comparison of publications to a disease was eradicated (pun intended).
Reviewer 2 Report
The article presents an important topic and is well written. There is a need for some editing for style and grammar.
There is one thing that needs to be mentioned/discussed in the Discussion. The author correctly says that the results reported in the article should be taken with a grain of salt. There is one more thing that should be mentioned. The stats could be a little different, if we took into account that it takes some time for Web of Science to cover the newest issue of a journal, and this could be due to their processes or the speed with which the publishers are submitting the journal content to them. It could well be that some of the articles that were published during the period of this study have not be covered by WoS. This might change the stats.
I think this is an interesting paper that is worth publishing.
Line 51: What is this number for?
Line 52: What is this number for?
Line 57: “systematic” rather than “systemic”
Line 84: Meaning is unclear, please reword this.
Line 87: please reword this section
Line 114: A sentence cannot begin with a number, please reword
Line 129: A sentence cannot begin with a number, please reword
Line 140-141: This section isn’t clear, please reword
Line 178-179: This section isn’t clear, please reword
Line 184: What is this number?
Line 197-199: This section isn’t clear, please reword
Line 201- 202: please ensure this is correct
Author Response
The article presents an important topic and is well written. There is a need for some editing for style and grammar.
Thank you very much for the support for our manuscript.
There is one thing that needs to be mentioned/discussed in the Discussion. The author correctly says that the results reported in the article should be taken with a grain of salt. There is one more thing that should be mentioned. The stats could be a little different, if we took into account that it takes some time for Web of Science to cover the newest issue of a journal, and this could be due to their processes or the speed with which the publishers are submitting the journal content to them. It could well be that some of the articles that were published during the period of this study have not be covered by WoS. This might change the stats.
There could have been papers that were published in this time frame on other coronaviruses, that we have missed by only consulting WoS. However, as the general trend of around 3-4 month to acceptance was true for the identified subset, we felt we do not need to dig deeper.
As for the papers on COVID-19, we have extended our search considerably by also including papers identified by the COVID-19 Open Research Dataset (CORD-19).
I think this is an interesting paper that is worth publishing.
Line 51: What is this number for?
Line 52: What is this number for?
All these numbers are times from submission to acceptance, or total publication times. As they are hard to read as a long list, I created a table out of them.
Line 57: “systematic” rather than “systemic”
Thank you very much! It is corrected
Line 84: Meaning is unclear, please reword this.
Line 87: please reword this section
As requested, the sentences have been reworded
Line 114: A sentence cannot begin with a number, please reword
The sentence has been changed to “Identifiable publication date was found for 97% of the documents (811/833)”
Line 129: A sentence cannot begin with a number, please reword
The sentence has been changed to “Publications were accepted on the very same day as they were submitted (received by the editorial office) in 5% (5/101) of the research papers, 11% (5/46) of the reviews, 8% (53) of the case studies, and 23% (38/167) of the other publications”
Line 140-141: This section isn’t clear, please reword
I tried to make it more understandable. While I give the data, I do not think this production time / lead lag means too much at this point, because journals put forward information even without much editing/production (i.e. provisional pdfs, accepted manuscripts, etc.). The revised part now reads: “This period could measure the speed of the production process, which transforms the accepted manuscript to a published article. But in many cases, provisional pdfs were posted, i.e. the actual production process has not been done. Furthermore, journals could also time their article publication.”
Line 178-179: This section isn’t clear, please reword
It has been changed to: “There might be one exception, in which data is withheld because the owners want to make sure their analysis of the data lands in a high impact factor journal. For example, in case of sequences, other scientist can also use them once they are shared with the scientific community, and potentially make the same analysis.”
Line 184: What is this number?
In my experience, this is the usual period for a grant.
Line 197-199: This section isn’t clear, please reword
It has been changed to: As another example, a paper claimed to observe an asymptomatic patient infecting others, but it turned out that the patient suppressed symptoms with medication and only appeared to be asymptomatic to outside observers, while feeling ill herself.
Line 201- 202: please ensure this is correct
This is exactly how it appeared in the original article that I cite. Please find it here: https://www.sciencemag.org/news/2020/02/paper-non-symptomatic-patient-transmitting-coronavirus-wrong
Reviewer 3 Report
Thanks for this timely contribution. I agreed to review it on the basis of the topic—and the time-sensitive nature of the content. The idea is excellent; the design is fine; the quantitative analysis is fine. Where the paper could be strengthened is simply in its presentation, particularly in the discussion, which feels a bit superficial, and at the sentence level. I am fully aware of the challenges of writing in a foreign language, but I get the feeling that the writing of this piece was a bit rushed. (In other fields, the irony of a piece—addressing the speed of a normally prolonged process—having a “rushed” feel would not be lost on me. But this piece is short enough that the extra effort invested in improving the English would be well worth it. Given the commentary on page 7 about the pressures to publish quickly resulting in errors, this piece should strive to be as clear and tight as possible.)
To start, I am not convinced that using the term “incubation period” in the title is in the best taste. (I could also point out that the transferability of the concept to writing would usually refer to the period in which the author is conceptualizing his or her ideas prior to writing a report or manuscript. In the time described in this piece, the author is simply waiting—all of his or her work has been completed by the time a manuscript is submitted. Really, then, you’re just referring to the “processing period”—which includes the manuscript-evaluation stage and everything else that needs to be completed before a journal submission can be made available for public consumption.)
Next, the abstract should be tighter. For example, the second sentence (lines 13–14) is too vague, particularly with the use of “around a year” and “seems to be.” The body of the piece is much more specific in this regard; why not introduce some of the statistics here. And, in lines 16–17, “these times shrunk by a magnitude” feels too unspecific to me, as well. What size magnitude?
The third sentence of the abstract introduces an opinion; but the abstract does not mention that any sort of discussion or assessment of the findings is also presented in the article. Perhaps a final sentence acknowledging that element would be helpful. (When I first read the abstract, in fact, my thought was that I’d learned everything already, since the results are so clearly presented in the final two sentences. I was wondering what more the manuscript would have to say—and about what.)
Lines 90–91; line 114; figure 1: It wasn’t completely clear to me, at first, whether the “editorials, news, commentary, perspectives, viewpoints, highlights” mentioned in section 2 (lines 90–91) are the types of materials included in the “other” category in section 3 (line 114; figure 1). It might be worth repeating what is included when the term “other” is introduced in section 3—or simply to note, when the list is offered in section 2, that these types of publications are collectively referred to as “other” in this study.
Line 165: “The first symptom of a paper is acceptance”: I’m not sure that symptom is the best word here.
Line 166: “Thus [the] incubation period is quite low for these papers”: I’m not sure that borrowing the concept of incubation period to reference a scholarly paper is the most tasteful here (same comment as with the title of the manuscript).
Line 174: If the “results need to be taken with a grain of salt,” then there’s no point in trying to publish this paper. Is it or is it not claiming to make a contribution? Actually, I disagree with the rationale proposed for why the results should be taken “with a grain of salt”: You’ve already made clear the time frame from which you captured your dataset. Of course you are considering only pieces that have appeared relatively quickly; your results are relevant to those cases and should not be dismissed. There is value in this piece—particularly because it puts figures on a phenomenon that people are aware is happening in the attempts to conquer COVID-19.
What about COVID-19 affecting journal operations? And are articles addressing COVID-19 receiving priority processing? (The only way to know, unless journals have said so themselves, would be to compare these results with a similar corpus of publications not related to COVID-19 that appeared in the same outlets within the same time frame. I’m not suggesting that the additional analysis needs to be conducted! You did make the astute—and helpful—comparison between papers on SARS and MERS that appeared in 2020, though it’s not clear how many papers you identified in the category of “other coronaviruses.” I would also, of course, be interested to know whether a similar speed-up in production times occurred with SARS and MERS—but that question would require additional analyses of different datasets from 2003 and 2012–13, and they do not need to be undertaken in order for this piece to make a contribution. Was any other research like the research presented here carried out on publications in the time of SARS or MERS? If so, a simple citation or two could solve my curiosity.)
Overall, this piece is timely and certainly interesting, but I’d like to see the language spruced up and perhaps a more robust discussion—one that emphasizes the role of scientific publications in defeating COVID-19 over the individualistic gains to be had by adding more lines to one’s CV through speedy publications.
Author Response
Thank you very much for your kind word on my manuscript. I tried to improve the presentation.
To start, I am not convinced that using the term “incubation period” in the title is in the best taste. (I could also point out that the transferability of the concept to writing would usually refer to the period in which the author is conceptualizing his or her ideas prior to writing a report or manuscript. In the time described in this piece, the author is simply waiting—all of his or her work has been completed by the time a manuscript is submitted. Really, then, you’re just referring to the “processing period”—which includes the manuscript-evaluation stage and everything else that needs to be completed before a journal submission can be made available for public consumption.)
While I thought that the tabloid mimicking title and the similarity of the language to the epidemic might enhance its style, I understand that it might be perceived as distasteful as scientific papers are compared to a disease, which was not my intention. These references are removed from the text.
Next, the abstract should be tighter. For example, the second sentence (lines 13–14) is too vague, particularly with the use of “around a year” and “seems to be.” The body of the piece is much more specific in this regard; why not introduce some of the statistics here. And, in lines 16–17, “these times shrunk by a magnitude” feels too unspecific to me, as well. What size magnitude?
There is considerable variation in the time it takes to publish and article from submission, and the available data suggest that most of this variation comes from the time from acceptance to publication. From the collected examples, I have changed this to 8–9 months.
The third sentence of the abstract introduces an opinion; but the abstract does not mention that any sort of discussion or assessment of the findings is also presented in the article. Perhaps a final sentence acknowledging that element would be helpful. (When I first read the abstract, in fact, my thought was that I’d learned everything already, since the results are so clearly presented in the final two sentences. I was wondering what more the manuscript would have to say—and about what.)
The abstract has been extended to cover my opinion on why this shortening appear.
Lines 90–91; line 114; figure 1: It wasn’t completely clear to me, at first, whether the “editorials, news, commentary, perspectives, viewpoints, highlights” mentioned in section 2 (lines 90–91) are the types of materials included in the “other” category in section 3 (line 114; figure 1). It might be worth repeating what is included when the term “other” is introduced in section 3—or simply to note, when the list is offered in section 2, that these types of publications are collectively referred to as “other” in this study.
This is now made explicit in the method and result section as well as in the figure caption.
Line 165: “The first symptom of a paper is acceptance”: I’m not sure that symptom is the best word here.
It has been changed to “The first milestone in the transformation of a manuscript to a paper is acceptance, at which time scientist might begin to brag about their new achievement.”
Line 166: “Thus [the] incubation period is quite low for these papers”: I’m not sure that borrowing the concept of incubation period to reference a scholarly paper is the most tasteful here (same comment as with the title of the manuscript).
Now it simply reads as “The time from submission to acceptance is quite low for these papers.”
Line 174: If the “results need to be taken with a grain of salt,” then there’s no point in trying to publish this paper. Is it or is it not claiming to make a contribution? Actually, I disagree with the rationale proposed for why the results should be taken “with a grain of salt”: You’ve already made clear the time frame from which you captured your dataset. Of course you are considering only pieces that have appeared relatively quickly; your results are relevant to those cases and should not be dismissed. There is value in this piece—particularly because it puts figures on a phenomenon that people are aware is happening in the attempts to conquer COVID-19.
One could object that papers already submitted, but undergoing considerably longer review would not be identified, and would skew the times to lower values. So, while I can and do claim that papers published in this time frame were peer-reviewed and published quickly, I cannot claim that it is true for all studies submitted in this time-frame. I changed the text slightly to make this point clearer.
What about COVID-19 affecting journal operations? And are articles addressing COVID-19 receiving priority processing? (The only way to know, unless journals have said so themselves, would be to compare these results with a similar corpus of publications not related to COVID-19 that appeared in the same outlets within the same time frame. I’m not suggesting that the additional analysis needs to be conducted! You did make the astute—and helpful—comparison between papers on SARS and MERS that appeared in 2020, though it’s not clear how many papers you identified in the category of “other coronaviruses.”
The search for “coronavirus” in WoS resulted in papers on SARS-CoV-2 (the focus of the study) and on other coronaviruses including SARS and MERS. All documents are listed in the supplementary table. Thus, the 64 documents omitted from the SARS-CoV-2/COVID-19 analysis, are all about other coronaviruses, including SARS and MERS.
I would also, of course, be interested to know whether a similar speed-up in production times occurred with SARS and MERS—but that question would require additional analyses of different datasets from 2003 and 2012–13, and they do not need to be undertaken in order for this piece to make a contribution. Was any other research like the research presented here carried out on publications in the time of SARS or MERS? If so, a simple citation or two could solve my curiosity.)
I have found no article presenting similar results for the other outbreaks. There were bibliometric studies showing that there was an increase in publication on coronaviruses after the SARS and also after the MERS outbreak, and similar trend were observed for the recent Ebola outbreak, and for the Zika outbreak in Central and South America. I have collected these observations in a new paragraph in the discussion.
Round 2
Reviewer 2 Report
The paper has been improved by adding new content and interpretation of results. There are a few grammar/spelling corrections that need to be addressed either by the author or the copyeditor. I am attaching the file with the edits that I have made.

Author Response
Thank you very much for correcting the grammatical mistakes I made. They are now corrected.